# Components of the Glutathione Cycle as Markers of Biological Age: An Approach to Clinical Application in Aging

**DOI:** 10.3390/antiox12081529

**Published:** 2023-07-30

**Authors:** Estefania Diaz-Del Cerro, Irene Martinez de Toda, Judith Félix, Adriana Baca, Monica De la Fuente

**Affiliations:** 1Unit of Animal Physiology, Department of Genetics, Physiology, and Microbiology, Faculty of Biological Sciences, Complutense University of Madrid, José Antonio Novais, 12, 28040 Madrid, Spain; imtcabeza@ucm.es (I.M.d.T.); jufelix@ucm.es (J.F.); adriabac@ucm.es (A.B.); mondelaf@bio.ucm.es (M.D.l.F.); 2Institute of Investigation Hospital 12 Octubre (imas12), 28041 Madrid, Spain

**Keywords:** aging, biological age, ImmunolAge, Immunity Clock, glutathione cycle

## Abstract

The oxidative-inflammatory theory of aging states that aging is the result of the establishment of a chronic oxidative-inflammatory stress situation in which the immune system is implicated. Among the redox parameters, those involved in the glutathione cycle have been suggested as essential in aging. Thus, the first objective of this study was to determine if several components of the glutathione cycle (glutathione reductase (GR) and glutathione peroxidase (GPx) activities, and concentrations of oxidized glutathione (GSSG) and reduced glutathione (GSH)) in leukocytes) are associated with the biological age (ImmunolAge) estimated using the Immunity Clock in 190 men and women. The second objective was to identify the best blood fraction (whole blood, blood cells, erythrocytes, or plasma) to quantify these components and correlate them with the estimated ImmunolAge. The results show that the oxidative state of peripheral leukocytes correlates with their functionality, supporting the idea that this is the basis of immunosenescence. In blood, the correlations are more significant in the fraction of blood cells with respect to ImmunolAge (positive correlations with GSSG concentration and the GSSG/GSH ratio, and negative correlations with GPx and GR activities). Therefore, blood cells are proposed as the most effective sample to estimate the biological age of individuals in clinical settings.

## 1. Introduction

Aging is a natural process characterized by a generalized deterioration of an organism’s functions, and the fact that the homeostatic systems (nervous, endocrine, and immune systems) are the most affected explains the age-related increased risks of morbidity and mortality [1]. The free radical theory of aging states that this process is the consequence of damage accumulation due to the deleterious oxidation of biomolecules caused by the high reactivity of free radicals and reactive oxygen species (ROS) produced in our cells because of the necessary use of oxygen. Accordingly, it has been demonstrated that aging is associated with an excess in ROS generation and a decrease in ROS removal ability via the defensive action of antioxidants, thereby generating a state of oxidative stress in the organism [1,2,3,4,5,6].

There are several antioxidant defenses that work together to keep the amounts of ROS within the appropriate levels necessary for their signaling roles. In this context, the glutathione cycle is one of the most important cycles that maintains the redox balance. Glutathione is a major intracellular antioxidant and redox regulator in cells. In addition to its essential role in redox homeostasis, it functions as a cofactor for a multitude of enzymes [7]. The glutathione peroxidase (GPx) enzyme uses reduced glutathione (GSH) as a cofactor to detoxify H_2_O_2_ into water and oxygen, generating oxidized glutathione (GSSG). Coupled with the action of GPx is the glutathione reductase (GR) enzyme. GR catalyzes the reduction of GSSG to GSH using NADPH as an electron donor. Thus, GR is responsible for maintaining the GSH/GSSG ratio by reducing GSSG generated during states of oxidative stress. The GSSG/GSH ratio is a sensitive indicator of changes in a cell’s thiol redox state and ongoing redox signaling, as well as the functional state of the cell [8].

Evidence from many studies with experimental animals and humans shows that glutathione concentrations are crucial in many biological functions and processes [9,10], but their concentrations decline with aging, leading to an increased pro-oxidizing shift and elevated oxidative stress [1,11,12,13,14,15,16,17,18,19,20]. Moreover, we have previously shown that the main components of the glutathione cycle, including GPx, GR, GSSG, and GSH, change with aging in both humans and mice and that a worse redox state in mice is related to a shorter lifespan [21]. These previous studies were performed by measuring the components of the glutathione cycle in immune cells in mice and in blood cells in humans. We chose to measure these levels in immune cells given the essential role that the immune system has in the aging process, being both a modulator and an accurate marker of the aging rate of an individual [1,22,23]. In fact, we have previously validated a mathematical model (Immunity Clock) based on immune function parameters (chemotaxis of neutrophils and lymphocytes, neutrophil phagocytosis, natural killer activity, and phytohemagglutinin-stimulated lymphoproliferation) to quantify the rate of aging or biological age in humans, which we have called ImmunolAge [24]. It is known that immune cells need a tightly regulated balance between oxidant and antioxidant compounds because they produce ROS to eliminate self- and non-self-damaging agents [1]. 

Nevertheless, it is still not known if the redox state in immune cells, and specifically several constituents of the glutathione cycle, correlates with specific innate and adaptive immune functions. In addition, for the quantification of redox markers in human blood, it is essential to know which fraction of the blood sample (whole blood, blood cells, erythrocytes, or plasma) is the golden one to reflect age-related trends in oxidative stress markers and the ImmunolAge of individuals. Thus, the first objective of this study is to examine whether the redox balance of peripheral leukocytes from the blood determines their function and correlates with the ImmunolAge quantified by the Immunity Clock in humans. Since isolating neutrophils and lymphocytes from human blood to estimate the ImmunolAge of each individual requires a time-consuming and laborious methodology that is difficult to apply in a clinical setting, we proposed measuring the components of the glutathione cycle in blood using a simple and affordable methodology for clinical applications. Because of that, the second objective is to identify the best blood fraction to measure the glutathione cycle parameters by investigating the correlation between these markers in different blood fractions (whole blood, blood cells, erythrocytes, or plasma) of the same individuals and their ImmunolAge.

## 2. Materials and Methods

### 2.1. Experimental Design

#### 2.1.1. Objective 1: To Verify Whether Oxidative Stress in Leukocytes is Associated with Immunosenescence

To realize this objective, several components of the glutathione cycle (activities of GPx and GR, concentrations of GSH and GSSG, and the GSSG/GSH ratio), as well as several functions (chemotaxis of neutrophils and lymphocytes, phagocytosis of neutrophils, NK antitumor cytotoxic activity, and lymphoproliferation in response to mitogens), were analyzed in leukocytes obtained from peripheral blood from humans (N = 40, 25 women and 15 men from 23 to 80 years old, with an average age of 45 ± 15 years). Then, we checked the correlation between these components and functions. Also, we investigated the correlation between the glutathione cycle components and the aging rate (ImmunolAge obtained by applying the Immunity Clock) of each man and woman. 

#### 2.1.2. Objective 2: To Identify the Best Blood Fraction to Measure the Glutathione Cycle Parameters

The components of the glutathione cycle were measured in whole blood, plasma, total blood cells, and isolated erythrocytes from the same individuals (N = 90, 45 women and 45 men from 23 to 80 years old, with an average age of 45 ± 15 years), and correlations with the ImmunolAge of each individual were performed. Consecutively, the results were validated in another cohort (N = 60, 30 women and 30 men from 23 to 80 years old, with an average age of 45 ± 15 years). 

### 2.2. Participants and Peripheral Blood Extraction

The participants were healthy men and women aged 23 to 80 years, from whom 12 mL of peripheral blood was extracted via a puncture of the antecubital vein. The blood was collected into tubes containing citrate, an anticoagulant. The exclusion criteria were as follows: (1) severe and unstable medical conditions or a history of chronic diseases; (2) psychiatric comorbidity; (3) taking medications, such as anti-inflammatory agents, muscle relaxants, corticoids, and antidepressants; (4) previous surgery; (5) pregnancy; (6) recent infections; and (7) non-cooperation during the evaluation The Human Ethics Committee of the 12 de Octubre Hospital (Madrid) approved this study (N° CEI 18/221).

### 2.3. Obtaining Different Blood Fractions: Whole Blood, Total Blood Cells, Isolated Erythrocytes, Plasma, and Leukocytes

Whole blood, plasma, and total blood cells: aliquots of 1 mL of different blood fractions were prepared. On the one hand, 1 mL of whole blood (without separating plasma from blood cells) was prepared. On the other hand, aliquots of whole blood were centrifuged at 1300× *g* for 20 min to separate the plasma from the blood cells. After centrifugation, the plasma was separated and preserved. Meanwhile, the pellets of total blood cells were reconstituted with RPMI 1640 (Ref. R8758, Sigma-Aldrich, Burlington, MA, USA). In all cases, aliquots of 100 μL were prepared and stored at −80 °C until the quantification of redox parameters.

Isolation of erythrocytes, neutrophils, and lymphocytes: We used 6 mL of blood for the isolation of erythrocytes, neutrophils, and lymphocytes using a Ficoll Histopaque density gradient of 1119 and 1077 g/cm^3^ (Ref. 11191 and 10771, Sigma-Aldrich, Burlington, MA, USA). The tubes were centrifuged at 700× *g* for 40 min. After centrifugation, erythrocytes were placed at the bottom of the tubes and separated from immune cells; they were collected using a Pasteur pipette, washed with PBS, used for preparing aliquots of 100 μL, and stored at −80 °C until analysis. The halos of neutrophils and lymphocytes were also collected using a Pasteur pipette, washed with PBS, and adjusted to the corresponding final concentrations for the development of each test while considering their viability (assessed using blue trypan, T8154, Sigma-Aldrich, Burlington, MA, USA).

### 2.4. Immune Function Tests

The phagocytic capacity of peripheral blood neutrophils in humans: This test was carried out following a method based on the ability of this cell type to ingest foreign bodies (latex balls, Ref. 59336, Sigma-Aldrich), which reflects their phagocytic function in vivo. Aliquots of 200 μL of neutrophil suspension were taken and incubated in migration inhibitory factor (MIF) plates for 30 min. Then, they were washed with PBS, and 20 μL of latex beads were added. After 30 min, they were fixed with methanol and stained with azure eosin methylene. The phagocytic index (P.I.: the number of latex particles ingested per 100 cells) and phagocytic efficacy (the percentage of cells capable of ingesting at least one latex particle) were determined using optical microscopy (×100) [22].

Chemotaxis ability of lymphocytes and phagocytes: Chemotaxis capacity is defined as the ability of cells (in this case, neutrophils or lymphocytes) to migrate or move in favor of a chemical gradient through chemoattractant signals released by the infectious focus. It was performed by modifying the Boyden method [22], which consists of a chamber with two compartments separated by a filter. In the compartment below, a 10^−8^ M formylated peptide (Fmlp, ref. F3506) is arranged as a chemoattractant. And in the upper compartment, the cellular suspension is added. After 3 h of incubation at 37°, the cells adhered to the filter (methanol at 50% and ethanol at 75%) and were fixed and stained (azure-eosin-methylene blue solution, GIEMSA, PANREAC). The chemotaxis index of neutrophils (C.I.N.) and lymphocytes (C.I.L.) was calculated, which is the number of cells counted in one-third of the lower face of the filter using light microscopy (×100), as previously described.

Natural killer activity (NK): This test was evaluated using a colorimetric assay (Cytotox 96 TM Promega, Boehringer Ingelheim, Ingelheim am Rhein, Germany) based on the determination of lactate dehydrogenase released by cytolysis of target cells (K562 cells) using tetrazolium salts. Effector cells (lymphocytes) and target cells were added in 96-well U-bottom culture dishes, with the ratio being 10:1. After 4 h of incubation, LDH activity was measured by adding the enzyme substrate and measuring absorbance at 490 nm. The results are expressed as the percentage of dead tumor cells (% lysis), as previously described [22].

Lymphoproliferative response (LP): This test was performed by measuring the incorporation of tritiated thymidine into cells, both at baseline and after stimulation with mitogen phytohemagglutinin (PHA, ref. 11082132001, Sigma-Aldrich), in 48-h cultures [22]. Briefly, leukocyte mononuclear suspensions were adjusted to 10^6^ lymphocytes/mL of gentamicin-supplemented RPMI and 10% fetal bovine serum, and 200 μL of this suspension was dispensed in 96-well plates, and 20 μL of PHA or RPMI (controls) was then added. After 48 h of incubation, 0.5 μCi/well of 3Htimidine was added (Dupont, Boston, MA, USA), followed by another 24 h of incubation. The results were calculated as the amount of 3H-thymidine (counts per minute, c.p.m.) under the baseline and stimulated conditions (PHA-LP) and expressed as lymphoproliferation capacity (%), given the value of 100 for the c.p.m. obtained under the baseline conditions, as previously described.

### 2.5. Calculation of ImmunolAge Using the Immunity Clock

After the analysis of the abovementioned immune functions for each subject, their ImmunolAge was calculated using the Immunity Clock, as previously described [24]. The mathematical formula is as follows:ImmunolAge: 93.943 − 0.23 × NK-0.001 × PHA-LP-0.022 × C.I.N.-0.02 × P.I.-0.019 × C.I.L.

### 2.6. Assessment of the Components of the Glutathione Cycle

Assessment of glutathione peroxidase (GPx) activity: The total activity of the enzyme was measured using cumene hydroperoxide (cumene-OOH, Ref. 820502, Sigma-Aldrich) as a substrate. This method determines the oxidation rate of glutathione (GSH) produced by cumene-OOH in a reaction catalyzed by the enzyme glutathione peroxidase, and it measures the decrease in absorbance at 340 nm due to the oxidation of NADPH (Ref. N7505, Sigma-Aldrich) in the presence of an excess of glutathione reductase (GR) enzyme [25]. For the calculation of GPx activity, which is expressed as mU/mg protein, the following equation was used:Activity GPx = ABS. catalyzed reaction − ABS. Uncatalyzed reaction
Total GPx activity (mU/mg proteins) = Act.GPx × F/E × X
where F is the dilution factor; E is the molar extinction coefficient of NADPH at 340 nm (6.22 × 10^−3^ M^−1^ cm^−1^); and X is mg of protein.

Assessment of glutathione reductase (GR) activity: This test was performed according to the technique based on the ability of GR to reduce oxidized glutathione. The measurement method is based on monitoring NADPH oxidation (a decrease in absorbance is observed over time), measured kinetically at 340 nm [26]. For the calculation of GR activity, which is expressed as mU/mg protein, the following equation was used:Total activity (mU/mg proteins) = ABSmin × Vt × FE × Vm × X
where Vt is the total volume in the cuvette (0.7 mL); F is the dilution factor; E is the molar extinction coefficient of NADPH at 340 nm (6.22 × 10^−3^ M^−1^ cm^−1^); Vm is the sample volume (0.05 mL); and X is mg of protein.

The concentration of oxidized glutathione (GSSG) and reduced glutathione (GSH): For the determination of GSH levels in the samples, the fluorescent probe O-phthaldialdehyde (OPT, ref. P1378; Sigma-Aldrich) was used. This method is based on the ability of reduced glutathione (GSH) to react with OPT at an optimal pH of 8, resulting in the formation of a fluorescent compound that is activated at 350 nm and presents a maximum emission point at 420 nm. The oxidized form of glutathione (GSSG) reacts optimally with OPT at a pH of 12. Since GSH is transformed into GSSG at a pH greater than 8, it is necessary to block this transformation with N-ethylmaleimide (NEM, Ref. 04259, Sigma-Aldrich) in order to quantify only the existing GSSG in the samples and not the result of oxidation of GSH when the medium is alkalized [21,27].

### 2.7. Statistical Analysis

The statistical analysis of the data was performed using the SPSS 22.0 program. To evaluate the interaction of ImmunolAge with the different redox and immune function parameters analyzed, correlation coefficients (r) were determined using Pearson’s test. Differences between the cases were considered significant when the *p*-value ≤ 0.05.

## 3. Results

### 3.1. Parameters of the Glutathione Cycle Correlate with the Functionality of Peripheral Neutrophils and with ImmunolAge

The results show that the values of the parameters of the glutathione cycle assessed in peripheral neutrophils of men and women correlate with those of their functions (Figure 1). In fact, the chemotaxis index correlates positively with GPx activity (r = 0.359, *p* = 0.035) but negatively with GSSG concentration (r = −0.413, *p* = 0.049). Moreover, the phagocytic index correlates positively with GSH concentration (r = 0.4368, *p* = 0.025) and negatively with GSSG concentration (r = −0.361, *p* = 0.048). As for ImmunolAge, most of the parameters of the glutathione cycle correlate with the ImmunolAge of men and women (Figure 1A–D). GPx (r = −0.458, *p* = 0.011) and GR activities (r = −0.373, *p* = 0.04), as well as GSH concentration (r = −0.4092, *p* = 0.03), correlate negatively with the ImmunolAge of each individual, while GSSG concentration correlates positively (r = 0.359, *p* = 0.039). 

### 3.2. Parameters of the Glutathione Cycle Correlate with the Functionality of Peripheral Lymphocytes and with ImmunolAge

In lymphocytes, the chemotaxis index correlates positively with GR activity (r = 0.399, *p* = 0.028) and GSH concentration (r = 0.358, *p* = 0.016); PHA-stimulated lymphoproliferation (r = −0.361, *p* = 0.038) and NK activity (r = −0.422, *p* = 0.006) correlate negatively with GPx activity, and NK activity correlates positively with GR activity (r = 0.369, *p* = 0.04). As for ImmunolAge, GR activity (r = −0.412, *p* = 0.019) and GSH concentration (r = −0.367, *p* = 0.025) correlate negatively with the ImmunolAge of each individual. The rest of Pearson’s correlations were not statistically significant (Figure 2I).

### 3.3. Blood Cells Are the Most Suitable Blood Fraction to Assess Parameters of the Glutathione Cycle to Estimate the Rate of Aging in Humans

To check which blood fraction best correlates with ImmunolAge in humans, the same parameters of the glutathione cycle were assessed in whole blood, blood cells, isolated erythrocytes, and plasma from the same individuals. The ImmunolAge of each individual was quantified using the Immunity Clock, and Pearson’s correlation coefficients were calculated between the redox markers and the ImmunolAge in each blood fraction.

Figure 3 shows the Pearson’s correlation coefficients obtained. In whole blood, GSSG concentration correlates positively (r = 0.231, *p* = 0.03) with the ImmunolAge of each individual. In the fraction of blood cells, GPx (r = −0.354, *p* = 0.001) and GR activities (r = −0.225, *p* = 0.050) correlate negatively with ImmunolAge, whereas GSSG (r = 0.243, *p* = 0.02) concentration and GSSG/GSH ratio (r = 0.251, *p* = 0.019) correlate positively with ImmunolAge. There are no statistically significant correlations with respect to isolated erythrocytes. Finally, in plasma, GPx activity (r = 0.257, *p* = 0.02) correlates negatively with the ImmunolAge of each individual, but GR activity (r = 0.264, *p* = 0.002), GSH concentration (r = 0.19, *p* = 0.087), and GSSG (r = 0.255, *p* = 0.01) concentration correlate positively with the ImmunolAge of each man and woman.

In another cohort, the same correlations between ImmunolAge and the glutathione cycle parameters analyzed in the different fractions of blood (blood cells, plasma, erythrocytes, and whole blood) were performed to validate the results obtained in the previous analysis. Similarly, in blood cells, GPx activity (r = −0.279, *p* = 0.03) correlates negatively, and GR activity (r = −0.272, *p* = 0.065) tends to correlate negatively with ImmunolAge, whereas GSSG concentration (r = 0.423, *p* = 0.009) and GSSG/GSH ratio (r = 0.452, *p* = 0.000) correlate positively with ImmunolAge (Figure 4). In the other blood fractions, no significant correlations were obtained, thus contrasting what we had observed previously.

## 4. Discussion

The aging population is increasing exponentially [28]. This fact makes it necessary to look for parameters that can be easily measured at the clinical level and allow the detection of the rate of aging of each person; in this case, the ImmunolAge, which is the biological age that we estimate using our validated Immunity Clock [24]. Thus, in cases where a person is identified as aging at a rapid rate, some strategy could be carried out to try to slow down the process. This would be of great importance for prevention and could be used to identify individuals at high risk of developing age-related diseases.

As a first objective, the present work investigated the relationship between the parameters of the glutathione cycle (GPx, GR, GSSG, and GSH) and several functions of peripheral blood leukocytes in humans. The results show that better immune functionality correlates with a better oxidative state (higher levels of GPx, GR, and GSH and lower values of GSSG and the GSSG/GSH ratio). These results are consistent with those obtained in previous studies in which it has been observed that elderly or prematurely aged individuals with impaired immune functions also have low GPx and GR activities, low GSH concentrations, and high levels of GSSG [1,21,29,30,31]. Thus, these results demonstrate that the states of these parameters are, at least in part, at the basis of the immunosenescence process that occurs with aging; the immune functions discussed in this article are crucial for health maintenance, and they are excellent markers of longevity and biological age (denominated ImmunolAge in this study since it is estimated by our Immunity Clock) [1,22,29] and can be used to calculate the rate of aging or ImmunolAge via the Immunity Clock, which was previously developed [24]. An analysis of the correlations between the glutathione cycle parameters in immune cells and ImmunolAge was also performed. The results show that these parameters of the glutathione cycle correlate with the ImmunolAge (negatively with GPx and GR activities and positively with GSSG) of each man and woman. These results suggest that these redox parameters could be used to quantify the rate of aging in humans in a simpler way than by measuring all the different immune functions included in the Immunity Clock.

Another objective of the present work was to identify which fraction of peripheral blood (whole blood, blood cells, erythrocytes, or plasma) is better for quantifying these parameters of the glutathione cycle. To realize this objective, GPx, GR, GSH, GSSG, and the GSSG/GSH ratio were measured in the different blood fractions from the same individuals, and an analysis of their correlations with ImmunolAge was performed.

Whole blood, consisting of plasma and blood cells (erythrocytes, leukocytes, and platelets), has the advantage of being more easily obtainable and reflecting the redox balance of other tissues, such as the liver, heart, and kidney [32,33]. However, only a statistically significant positive correlation was obtained between ImmunolAge and GSSG concentration. The measurement of GSSG in whole blood is considered a useful indicator of the whole oxidative state and disease risk in humans [34], and it has been demonstrated that GSSG concentration in whole blood increases with aging [19]. 

Regarding blood cells (erythrocytes and leukocytes), obtained via centrifugation of whole blood and removal of plasma, negative correlations were obtained between two antioxidant defenses, GPx and GR activities, and ImmunolAge, and positive correlations were obtained between the concentration of GSSG and the GSSG/GSH ratio and ImmunolAge. These results coincide with those obtained in previous studies, where a decrease in these antioxidant activities was observed with advancing age in total blood cells [19,35,36]. In this way, individuals who have a lower GPx activity, a higher GSSG concentration, and a higher GSSG/GSH ratio probably have decreased immune functions and, therefore, obtain a higher biological age, as calculated by the “Immunity Clock” [24]. However, other markers of biological age estimation have also been linked to increased oxidative stress in blood cells. In fact, it has been observed that an increase in ROS can generate alterations in the DNA methylation pattern and, consequently, increase the biological age calculated via the “Epigenetic Clock” [37]. Other biological clocks, such as telomere shortening, have also been shown to be affected by oxidative stress [38], and several studies have shown that shorter telomere length is associated with older age [39].

As for erythrocytes, they are known to be the most common type of blood cells and are crucial for the delivery of oxygen in vertebrates [40]. In addition, when transporting oxygen, they are highly exposed to oxidation, so they have an effective antioxidant protective system [41]. However, no statistically significant correlations were obtained between the glutathione cycle parameters and ImmunolAge. Since erythrocytes, as with whole blood, show hardly any significant correlations with respect to the parameters analyzed, the hypothesis is that it is circulating leukocytes that suffer more evident variations in their redox state when the speed of aging is accelerated [22,23]. Their important role in the rate of aging seems hard to reconcile with the widely held view that immune cells, such as neutrophils, lymphocytes, and NK cells, are very short-lived, with a circulatory half-life of 7 h, 7 days, and 10 days, respectively [42,43,44]. In fact, the half-life of each leukocyte subpopulation does not necessarily influence the role that these cells play in the rate of aging. However, new techniques need to be applied to accurately monitor immune cells’ half-lives in order to study their link with the biological age of individuals.

Finally, in plasma, although ImmunolAge correlates negatively with GPx activity, which makes sense since the activity of this enzyme decreases significantly in plasma with advancing age [45,46,47], GR activity correlates positively with ImmunolAge. This demonstrates that there are not only differences between studies due to the types of samples but there may also be apparent inconsistencies between parameters that are analyzed in the different samples in the same study. With advancing age, there is an increase in GSSG, which could explain the increase in GR acting as a compensatory mechanism. However, another possibility is the release of GR by blood cells into plasma since this enzymatic activity is fundamentally intracellular. Regarding oxidative compounds, such as GSSG, a positive correlation with ImmunolAge was obtained, which is in line with previous studies [48]. Therefore, we conclude that plasma is not the best type of sample to assess parameters that can serve as markers of biological age since it does not follow the normal pattern of aging.

In addition to glutathione, there are many other SH-dependent groups that are highly involved in aging and are very important for longevity, such as peroxiredoxin or glutaredoxin [49,50,51,52]. However, they were not investigated in this work, which is a limitation of this study. Therefore, it would be very interesting to study these other SH-molecules and their relationships with the biological age of individuals to better understand the links between SH-dependent groups and the rate of aging in order to find new biomarkers of biological age.

## 5. Conclusions

In conclusion, it can be argued that total blood cells provide the best sample to assess the states of these parameters of the glutathione cycle during the aging process and to estimate the biological age of each individual in a clinical setting. This is because total blood cells are the only blood fraction that reflects the usual age-related behavior of the parameters described in previous studies, such as the decrease in antioxidant enzymes (GPx and GR activities) and increases in oxidative compounds (GSSG and GSSG/GSH ratio). In addition, this is the blood sample that displays the highest significant correlations with the ImmunolAge of the participants in this study. Although total blood cells are not as easy to obtain as whole blood, the fact that this blood fraction’s redox state correlates with the ImmunolAge of individuals suggests that the glutathione cycle parameters analyzed in blood cells could be used to quantify the rate of aging in humans. The quantification of these parameters could be easily measured in any clinical setting, which would help in the identification of individuals aging at a fast rate and, consequently, with a higher risk of morbidity and mortality. These individuals could undergo lifestyle changes or specific anti-aging interventions to help the aging population achieve healthy aging.

## Figures and Tables

**Figure 1 antioxidants-12-01529-f001:**
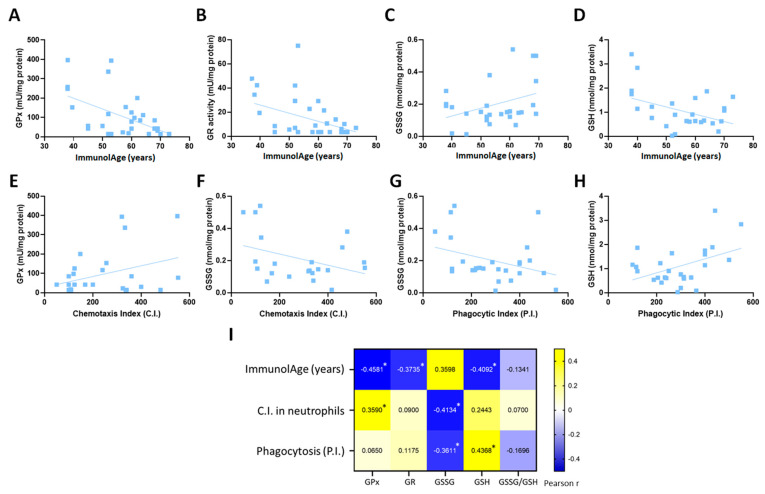
Assessment of parameters of the glutathione cycle and immune functionality of peripheral neutrophils in humans: (**A**–**D**) correlations of the glutathione cycle with ImmunolAge; (**E**–**H**) correlations of parameters of the glutathione cycle with neutrophil functions; and (**I**) heatmap of the Pearson’s correlation coefficients among ImmunolAge, neutrophil functions, and parameters of the glutathione cycle (* *p* < 0.05). C.I.: chemotaxis index; P.I.: phagocytic index; GPx: glutathione peroxidase; GR: glutathione reductase; GSSG: oxidized glutathione; GSH: reduced glutathione.

**Figure 2 antioxidants-12-01529-f002:**
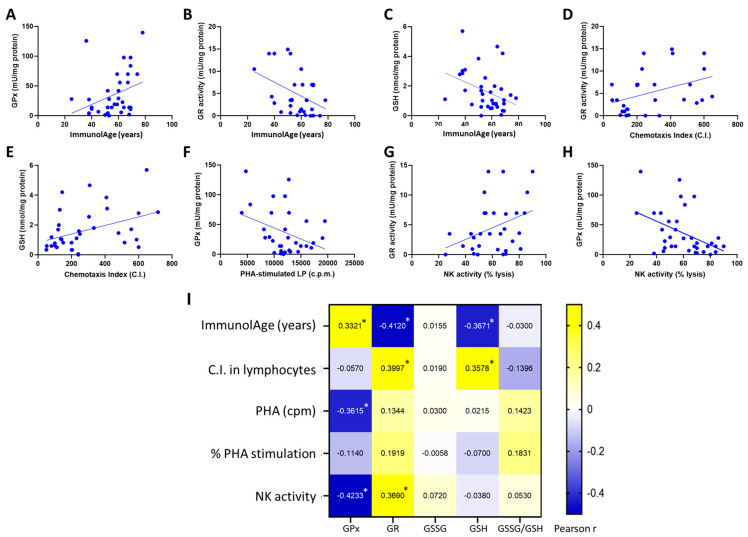
Assessment of parameters of the glutathione cycle and immune functionality of peripheral lymphocytes in humans: (**A**–**C**) correlations of the glutathione cycle with ImmunolAge; (**D**–**H**) correlations of parameters of the glutathione cycle with lymphocyte functions; and (**I**) heatmap of the Pearson’s correlation coefficients among ImmunolAge, lymphocyte functions, and parameters of the glutathione cycle (* *p* < 0.05). C.I.: chemotaxis index; NK: natural killer; LP: lymphoproliferation; PHA: phytohemagglutinin; c.p.m.: counts per minute; GPx: glutathione peroxidase; GR: glutathione reductase; GSSG: oxidized glutathione; GSH: reduced glutathione.

**Figure 3 antioxidants-12-01529-f003:**
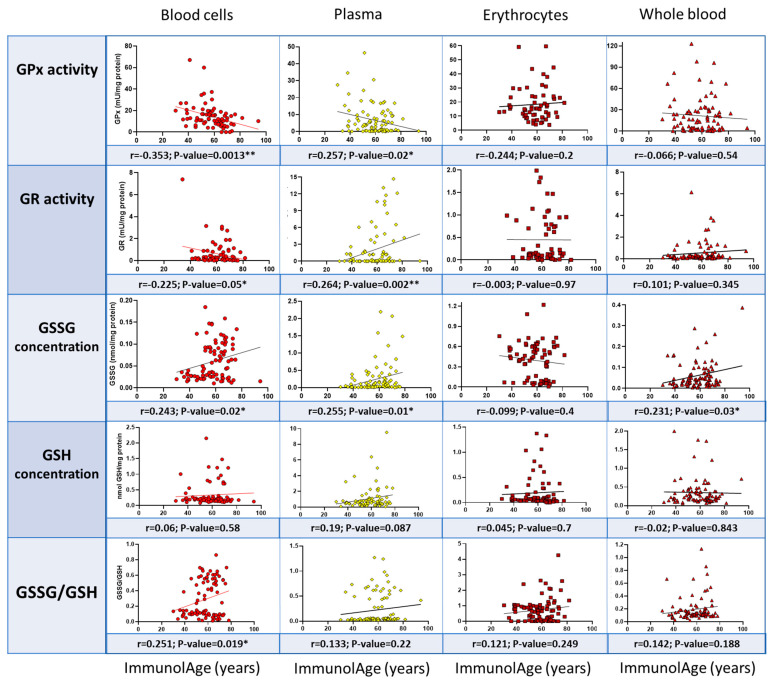
Pearson’s correlations between ImmunolAge and parameters of the glutathione cycle in different blood fractions of humans. GPx: glutathione peroxidase; GR: glutathione reductase; GSSG: oxidized glutathione; GSH: reduced glutathione. * *p* < 0.05 and ** *p* < 0.01.

**Figure 4 antioxidants-12-01529-f004:**
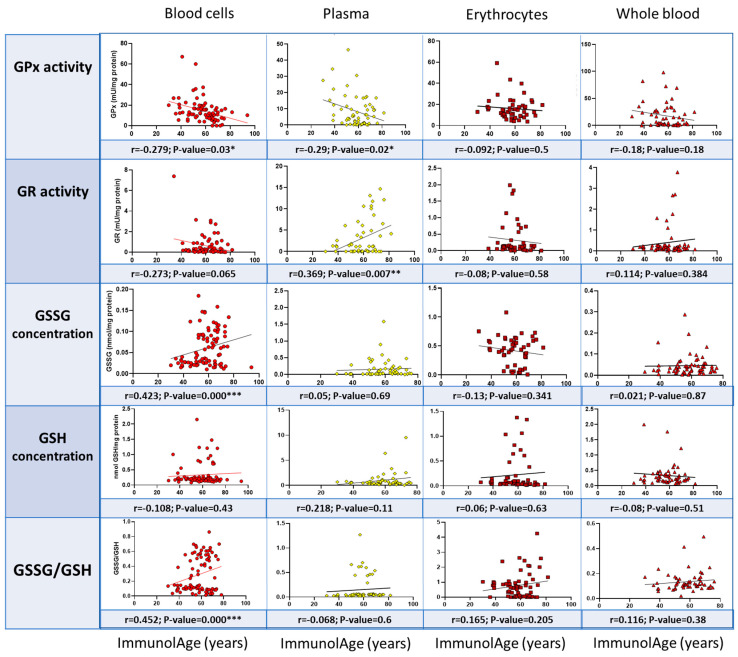
Validation of the results in different blood fractions. Pearson’s correlations between ImmunolAge and parameters of the glutathione cycle in another cohort of men and women. * *p* < 0.05, ** *p* < 0.01, and *** *p* < 0.001.

## Data Availability

Data is contained within the article.

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
