# Peer review of "Components of the Glutathione Cycle as Markers of Biological Age: An Approach to Clinical Application in Aging"

_antioxidants, 2023, doi:10.3390/antiox12081529_

Round 1

Reviewer 1 Report

This study was carried out to verify that the oxidative stress of leukocytes is associated with immunosenescence using healthy participants aged between 23 and 80. The objective is interesting and important for applying the knowledge in clinical settings. However, unfortunately, manuscript preparation is sloppy and hard to follow.

Major issues

1, Approved identification number of the Human Ethics Committee should be added in the text.

2, In Fig1a,b there are no data points for the participants younger than 40. Why?

3, Fig,1 and the legend do not match completely.

4. In Fig 2 there is no LP proliferation data, although LP is shown in the figure legend.

5, It is very hard to follow the text; for example, in line 186, it is stated  “In the neutrophils, the chemotaxis index correlates positively with GPx activity (R2= 0.1289, P=0.035)”. However, there is no correspondent data (R2= 0.1289, P=0.035) in all figures. The authors must make the presentation much clearer and easier to let readers understand.

Minor issues

1.       Line 52, H2O2 should ameliorate the expression.

2, Abovementioned should read as above mentioned.

Needs extensive editing.

Author Response

Reviewer 1: This study was carried out to verify that the oxidative stress of leukocytes is associated with immunosenescence using healthy participants aged between 23 and 80. The objective is interesting and important for applying the knowledge in clinical settings. However, unfortunately, manuscript preparation is sloppy and hard to follow.

Major issues

  1. Approved identification number of the Human Ethics Committee should be added in the text.

We thank the reviewer for notifying us of the absence of the approval number from the ethics committee, which we have added in the revised version of the manuscript: “This study was conducted according to the guidelines of the Declaration of Helsinki and approved by the Ethics Committee of the Hospital 12 de Octubre of Madrid (N° CEI 18/221)”.

  1. In Fig1a,b there are no data points for the participants younger than 40. Why?
  2. Fig,1 and the legend do not match completely.

We thank the reviewer for notifying us of these mistakes. The legend has been corrected, and the data points have been included in Fig 1A and B for participants with a biological age younger than 40.

  1. In Fig 2 there is no LP proliferation data, although LP is shown in the figure legend.

We apologize for the confusion, but in Figure 2, there are lymphoproliferation (LP) data, which are displayed in Fig 2F and in the PHA (c.p.m.) and % PHA stimulation rows of the heatmap.

  1. It is very hard to follow the text; for example, in line 186, it is stated “In the neutrophils, the chemotaxis index correlates positively with GPx activity (R2= 0.1289, P=0.035)”. However, there is no correspondent data (R2= 0.1289, P=0.035) in all figures. The authors must make the presentation much clearer and easier to let readers understand.

We thank the reviewer for the comment. Following the suggestion, we have changed the Results section in order to make it easier for readers to read and understand.

Minor issues

  1. Line 52, H2O2 should ameliorate the expression.
  2. Abovementioned should read as above mentioned.

We thank the reviewer for notifying us of these errors, which have been corrected in the revised version of the manuscript.

Reviewer 2 Report

Cerro et al. studied here the correlations between glutathione cycle elements (substrates and enzymes) in leukocytes and the biological age of volunteers estimated by the Immunity Clock.

The MS was reasonably well-written (English Language needs to be rechecked, thought), but I particularly dislike the two-objective approach of the study. From my point-of-view, the authors should look for an integration of both approaches or simply limit this study for a single one. If the scientific deep of the singles studies (after split) is not worthy for a pubication, that is another question. Moreover, the topic lacks novelty, since the interdependence between GSH/GSSG levels and aging, in general aspects, is relatively well-established. The MS needs an innovative approach and substantial improvement before acceptance for publication, from my point of view.

I have some serious concerns about the MS as it stands:

(i) it is well-known that thiol(SH)-dependent groups are the most redox senstivie moieties in biological systems; therefore, these indexes are relatively accurate to predict the redox scenario within cells, tissues etc. On the other hand, many other SH-dependent groups are tightly  involved in sensiting oxidative challenges in vivo: e.g. peroxiredoxin, glutaredoxins, etc. From my point of view, it is inevitable to mention and to monitor these indexes in order to obtain a more accurate redox panel in blood, plasma, or circulating immune cells. If the authors are unable to include these measurements in their study, at least a robust discussion about the role of these SH-molecules in aging is necessary.

REFS:.

Ismail T, Kim Y, Lee H, Lee DS, Lee HS. Interplay Between Mitochondrial Peroxiredoxins and ROS in Cancer Development and Progression. Int J Mol Sci. 2019 Sep 7;20(18):4407. doi: 10.3390/ijms20184407. PMID: 31500275; PMCID: PMC6770548.

Detienne G, De Haes W, Mergan L, Edwards SL, Temmerman L, Van Bael S. Beyond ROS clearance: Peroxiredoxins in stress signaling and aging. Ageing Res Rev. 2018 Jul;44:33-48. doi: 10.1016/j.arr.2018.03.005. Epub 2018 Mar 23. PMID: 29580920.

Oláhová M, Taylor SR, Khazaipoul S, Wang J, Morgan BA, Matsumoto K, Blackwell TK, Veal EA. A redox-sensitive peroxiredoxin that is important for longevity has tissue- and stress-specific roles in stress resistance. Proc Natl Acad Sci U S A. 2008 Dec 16;105(50):19839-44. doi: 10.1073/pnas.0805507105. Epub 2008 Dec 8. PMID: 19064914; PMCID: PMC2604961.  

Xiao Z, La Fontaine S, Bush AI, Wedd AG. Molecular Mechanisms of Glutaredoxin Enzymes: Versatile Hubs for Thiol-Disulfide Exchange between Protein Thiols and Glutathione. J Mol Biol. 2019 Jan 18;431(2):158-177. doi: 10.1016/j.jmb.2018.12.006. Epub 2018 Dec 12. PMID: 30552876.

(ii) The authors need to asure (I believe with additional experiments) that the isolated neutrophils and lymphocytes were not previously activated (during iolation procedures), a condition that increases their NADPH oxidase activity (NOX). The assembling of NOX unit in the immune cell membrane abruptally increases the production of ROS/RNS that would significantly affect GSH, GSSG, and GSH/GSSG ratio in samples. For example: measure GSH, GSSG contents after immune cell activation with phorbol-12-myristate acetate (PMA), as a background maximum scale.

(iii) The discussion section should also discuss the half-lives of each circulating immune cell and their effect on human longevity

REF:. 

Tak T, Tesselaar K, Pillay J, Borghans JA, Koenderman L. What's your age again? Determination of human neutrophil half-lives revisited. J Leukoc Biol. 2013 Oct;94(4):595-601. doi: 10.1189/jlb.1112571. Epub 2013 Apr 26. PMID: 23625199.

(iv) please, include an illustration to clearly show the interplaybetween  antioxidant SH-groups and redox balance within cells and tissues

REF:. 

Barros MP, Rodrigo MJ, Zacarias L. Dietary Carotenoid Roles in Redox Homeostasis and Human Health. J Agric Food Chem. 2018 Jun 13;66(23):5733-5740. doi: 10.1021/acs.jafc.8b00866. Epub 2018 Jun 5. PMID: 29785849.

Iskusnykh IY, Zakharova AA, Pathak D. Glutathione in Brain Disorders and Aging. Molecules. 2022 Jan 5;27(1):324. doi: 10.3390/molecules27010324. PMID: 35011559; PMCID: PMC8746815.

(iii)  

From my point of view, the MS needs a full English revision, as many typos, grammar issues were here detected.

Author Response

Reviewer 2: Cerro et al. studied here the correlations between glutathione cycle elements (substrates and enzymes) in leukocytes and the biological age of volunteers estimated by the Immunity Clock.

The MS was reasonably well-written (English Language needs to be rechecked, thought), but I particularly dislike the two-objective approach of the study. From my point-of-view, the authors should look for an integration of both approaches or simply limit this study for a single one. If the scientific deep of the singles studies (after split) is not worthy for a publication, that is another question. Moreover, the topic lacks novelty, since the interdependence between GSH/GSSG levels and aging, in general aspects, is relatively well-established. The MS needs an innovative approach and substantial improvement before acceptance for publication, from my point of view.

We understand the reviewer’s point of view, but, in our opinion, the results presented in this manuscript are novel for several reasons. First, we believe that the two objectives of the study should be presented in the same publication because they show two sides of the same coin. On the one hand, the first objective shows that glutathione cycle parameters are excellent markers of aging rate (biological age) when measured in leukocytes. However, obtaining leukocytes is too expensive and laborious to be performed routinely in any clinical laboratory. Therefore, it is necessary to look for another blood sample that is more easily obtained and in which the relationship of these parameters with biological age can continue to be observed. This is the impetus for carrying out the second objective of the study. It is true that the parameters analyzed in this study on aging have been studied a lot; however, there is much controversy in the results obtained in previous studies depending on the study population, methodology, and type of samples used. Therefore, an important strength of our study is that different types of "samples" (blood, blood cells, isolated erythrocytes, plasma, and leukocytes) were obtained from the same participants and, thus, the results can be directly compared, and clearer conclusions can be drawn.

I have some serious concerns about the MS as it stands:

(i) it is well-known that thiol(SH)-dependent groups are the most redox sensitive moieties in biological systems; therefore, these indexes are relatively accurate to predict the redox scenario within cells, tissues etc. On the other hand, many other SH-dependent groups are tightly involved in sensiting oxidative challenges in vivo: e.g. peroxiredoxin, glutaredoxins, etc. From my point of view, it is inevitable to mention and to monitor these indexes in order to obtain a more accurate redox panel in blood, plasma, or circulating immune cells. If the authors are unable to include these measurements in their study, at least a robust discussion about the role of these SH-molecules in aging is necessary.

REFS:.

Ismail T, Kim Y, Lee H, Lee DS, Lee HS. Interplay Between Mitochondrial Peroxiredoxins and ROS in Cancer Development and Progression. Int J Mol Sci. 2019 Sep 7;20(18):4407. doi: 10.3390/ijms20184407. PMID: 31500275; PMCID: PMC6770548.

Detienne G, De Haes W, Mergan L, Edwards SL, Temmerman L, Van Bael S. Beyond ROS clearance: Peroxiredoxins in stress signaling and aging. Ageing Res Rev. 2018 Jul;44:33-48. doi: 10.1016/j.arr.2018.03.005. Epub 2018 Mar 23. PMID: 29580920.

Oláhová M, Taylor SR, Khazaipoul S, Wang J, Morgan BA, Matsumoto K, Blackwell TK, Veal EA. A redox-sensitive peroxiredoxin that is important for longevity has tissue- and stress-specific roles in stress resistance. Proc Natl Acad Sci U S A. 2008 Dec 16;105(50):19839-44. doi: 10.1073/pnas.0805507105. Epub 2008 Dec 8. PMID: 19064914; PMCID: PMC2604961.  

Xiao Z, La Fontaine S, Bush AI, Wedd AG. Molecular Mechanisms of Glutaredoxin Enzymes: Versatile Hubs for Thiol-Disulfide Exchange between Protein Thiols and Glutathione. J Mol Biol. 2019 Jan 18;431(2):158-177. doi: 10.1016/j.jmb.2018.12.006. Epub 2018 Dec 12. PMID: 30552876.

We thank the reviewer for the comment and suggestion. We agree with the reviewer that there are many other SH-molecules and oxidative compounds that have an important role in aging. However, in our study, we measured the enzymatic activities of glutathione peroxidase and glutathione reductase, the concentrations of reduced glutathione (GSH) and oxidized glutathione (GSSG), and the GSSG/GSH ratio (and not others) because our research group has extensive experience analyzing these compounds and has observed in multiple studies how they vary with chronological aging and in models of premature/accelerated aging;  we have shown that these oxidative parameters are excellent markers of longevity and aging speed when measured in leukocytes. Thus, we wanted to gain further in-depth knowledge of the link between the state of the glutathione cycle parameters and the immune function in the aging process as well as what type of sample is the most appropriate and easiest to obtain to quantify these parameters. However, following the Reviewer’s suggestion, we have added this aspect as a limitation of the study, which will be studied in future projects: “In addition to glutathione, there are many other SH-dependent groups that are highly involved in aging and are very important for longevity, such as peroxiredoxin or glutaredoxins [46-49]. However, they were not investigated in this study, which is a limitation of this study. Therefore, it would be very interesting to study these other SH-molecules and their relationships with the biological age of individuals to better understand the links between SH-dependent groups and the rate of aging in order to find new biomarkers of biological age.”

(ii) The authors need to asure (I believe with additional experiments) that the isolated neutrophils and lymphocytes were not previously activated (during iolation procedures), a condition that increases their NADPH oxidase activity (NOX). The assembling of NOX unit in the immune cell membrane abruptally increases the production of ROS/RNS that would significantly affect GSH, GSSG, and GSH/GSSG ratio in samples. For example: measure GSH, GSSG contents after immune cell activation with phorbol-12-myristate acetate (PMA), as a background maximum scale.

First, we would like to thank the reviewer for the comment. We agree that any experimental procedure that involves extracting and isolating cells from their natural environment will stress them out; therefore, the values of functionality that we observed would not be 100% comparable to what we would observe in an organism in real-life conditions. However, all the samples were treated in the same way and under the same conditions, so that the values could be compared without triggering the error that some cells are more activated than others. In addition, we have published numerous studies in which we had used the same techniques as in this study and in which we had validated the analysis of immune functions and oxidative parameters presented in this article (chemotaxis, phagocytosis, natural killer activity, lymphoproliferation, glutathione peroxidase, and reductase activity, and GSSG and GSH concentration) as predictors of longevity and biological age (Martínez de Toda et al., 2016, 2019, 2020, 2020b, 2021).

Martínez de Toda I, Vida C, Díaz-Del Cerro E, De la Fuente M. The Immunity Clock. J Gerontol A Biol Sci Med Sci. 2021;76(11):1939-1945. doi:10.1093/gerona/glab136

Martínez de Toda I, Vida C, Garrido A, De la Fuente M. Redox Parameters as Markers of the Rate of Aging and Predictors of Life Span. J Gerontol A Biol Sci Med Sci. 2020;75(4):613-620. doi:10.1093/gerona/glz033

Martínez de Toda I, Maté I, Vida C, Cruces J, De la Fuente M. Immune function parameters as markers of biological age and predictors of longevity. Aging (Albany NY). 2016;8(11):3110-3119. doi:10.18632/aging.101116

Martínez de Toda I, Vida C, Sanz San Miguel L, De la Fuente M. Function, Oxidative, and Inflammatory Stress Parameters in Immune Cells as Predictive Markers of Lifespan throughout Aging. Oxid Med Cell Longev. 2019;2019:4574276. Published 2019 Jun 2. doi:10.1155/2019/4574276

Martínez De Toda I, Vida C, García-Salmones M, Alonso-Fernández P, De La Fuente M. Immune Function, Oxidative, and Inflammatory Markers in Centenarians as Potential Predictors of Survival and Indicators of Recovery After Hospital Admission. J Gerontol A Biol Sci Med Sci. 2020b;75(10):1827-1833. doi:10.1093/gerona/glz250

 (iii) The discussion section should also discuss the half-lives of each circulating immune cell and their effect on human longevity

REF:

Tak T, Tesselaar K, Pillay J, Borghans JA, Koenderman L. What's your age again? Determination of human neutrophil half-lives revisited. J Leukoc Biol. 2013 Oct;94(4):595-601. doi: 10.1189/jlb.1112571. Epub 2013 Apr 26. PMID: 23625199.

We find the reviewer’s comment very interesting, so we have added in the Discussion section the following information discussing the possible role of the half-lives of immune cells in biological age: “…Their important role in the rate of aging seems hard to reconcile with the widely held view that immune cells, such as neutrophils, lymphocytes, and NK cells, are very short-lived, with a circulatory half-life of 7 h, 7 days, 10 days respectively (Elson et al., 1976; Tak et al. 2013; Nayar et al., 2015). In fact, the half-life of each leukocyte subpopulation does not necessarily influence the role that these cells play in the rate of aging. However, new techniques need to be applied to accurately monitor immune cells’ half-lives in order to study their link with the biological age of individuals.”

-Nayar, S., Dasgupta, P., Galustian, C. Extending the lifespan and efficacies of immune cells used in adoptive transfer for cancer immunotherapies-A review. Oncoimmunology 2015, 19;4(4):e1002720.

-Elson, C.J., Jablonska, K.F., Taylor, R.B. Functional half-life of virgin and primed B lymphocytes. Eur J Immunol 1976, 6(9):634-638.

(iv) please, include an illustration to clearly show the interplay between antioxidant SH-groups and redox balance within cells and tissues

REF:

Barros MP, Rodrigo MJ, Zacarias L. Dietary Carotenoid Roles in Redox Homeostasis and Human Health. J Agric Food Chem. 2018 Jun 13;66(23):5733-5740. doi: 10.1021/acs.jafc.8b00866. Epub 2018 Jun 5. PMID: 29785849.

Iskusnykh IY, Zakharova AA, Pathak D. Glutathione in Brain Disorders and Aging. Molecules. 2022 Jan 5;27(1):324. doi: 10.3390/molecules27010324. PMID: 35011559; PMCID: PMC8746815.

Following the reviewer's recommendation, we have added a new graphical abstract to the revised version of the manuscript.

Reviewer 3 Report

This paper is an ambitious attempt to determine biological age from peripheral samples, and I read it with great interest. It was interesting, but the writing style was a bit clunky and difficult to read in many places. I may be mistaken, but I would like to point out a possible problem with the statistical calculations. Below are some of my points of concern, which I hope will be useful in the revision process.

1. I think it is a significant study. However, I think it is difficult to approach because of the use of the unfamiliar term ImmnolAge and the monotonous nature of the figures, which are difficult to understand what they are doing at a quick glance. Although it is not required, I think this paper would be more interesting if there were something like a graphical abstract that would give an idea of the contents of this paper at a glance.

2. Methods should be explained in more detail rather than just citing their own paper. Especially 2.3 and 2.4 should have more details.

3. Related to the above, I have the impression that many of the definitions of terms are vague; more explanation should be given in the Introduction regarding Immunological clock and ImmunolAge. Please keep in mind that the readability of the paper should be somewhat understandable without having read the previous paper.

4. I think Biological age and ImmunolAge are used as synonymous in this paper, but it is confusing when they are mixed in the paper and in the graphs. Biological age should be able to be defined from various perspectives not only immunological functions, so why not unify it with ImmunolAge in this paper?

5. I think there is an approval number from the ethics review committee, which should also be indicated.

6. I was wondering if there is any bias in the way the specimens were collected. This should be a little more descriptive here. Is there a bias toward certain age groups? Also, wouldn't the values vary considerably depending on whether the person has had a severe infection or not? I would be curious to know if such people are separated. Of course, the sample of people is limited, so it may not be possible to go that far, but in that case, I think it should be described as a limitation of this paper.

7. The authors use the notation P=0.05 in the text, but a significant difference usually occurs when P is less than a specified value, such as 0.05>P, so this notation could be seen as no difference. Looking at the statistics section, it fits the authors' definition, but it is different from the definitions in other papers, which makes me feel uncomfortable. I would like them to explain the justification for this definition.

8. I don't deny the possibility that P=0.05, but it has been mentioned many times in this paper. I think the probability is low, and I don't mean to be rude, but I would appreciate your scrutiny to see if this is not some kind of error.

9. For the purpose of this issue, is it necessary to look at women and men inclusively? I thought the values might be quite different by gender. A close examination of the paper reveals a mention of Gender in Lines 246-247, but this may be an overstatement since there are no data analyzed separately for men and women.

10. Many figures have many samples with vertical axis values close to 0. Can they rule out the possibility that these are not measured due to sample problems? If there are criteria for evaluation, they should be stated in the method.

11. There are two H's in Figure 1.

12. If I am not mistaken, the explanations for Fig. 1 and Fig. 2 are mixed in 3.1 or there is no explanation for Fig. 2 in the text? Clarifying whether the description refers to Figure 1 or Figure 2 would increase readability.

Author Response

Reviewer 3: This paper is an ambitious attempt to determine biological age from peripheral samples, and I read it with great interest. It was interesting, but the writing style was a bit clunky and difficult to read in many places. I may be mistaken, but I would like to point out a possible problem with the statistical calculations. Below are some of my points of concern, which I hope will be useful in the revision process.

  1. I think it is a significant study. However, I think it is difficult to approach because of the use of the unfamiliar term ImmnolAge and the monotonous nature of the figures, which are difficult to understand what they are doing at a quick glance. Although it is not required, I think this paper would be more interesting if there were something like a graphical abstract that would give an idea of the contents of this paper at a glance.

We welcome the comments of the reviewer as they have helped us see that an image could help readers quickly understand the main ideas of the article. Therefore, we have added a graphical abstract to the revised version of the manuscript.

  1. Methods should be explained in more detail rather than just citing their own paper. Especially 2.3 and 2.4 should have more details.

We thank the reviewer's comment. Following the suggestion, we have expanded some aspects of the “Material and Methods” section.

  1. Related to the above, I have the impression that many of the definitions of terms are vague; more explanation should be given in the Introduction regarding Immunological clock and ImmunolAge. Please keep in mind that the readability of the paper should be somewhat understandable without having read the previous paper.

We thank the reviewer for the suggestion and have taken them into account in the new version of the manuscript.

  1. I think Biological age and ImmunolAge are used as synonymous in this paper, but it is confusing when they are mixed in the paper and in the graphs. Biological age should be able to be defined from various perspectives not only immunological functions, so why not unify it with ImmunolAge in this paper?

We thank the reviewer for the comment and agree that it can be confusing for readers to use both biological age and ImmunolAge as synonymous. Thus, we have unified the terms as ImmunolAge in the revised manuscript.

  1. I think there is an approval number from the ethics review committee, which should also be indicated.

We thank the reviewer for noticing the absence of the approval number from the ethics committee, which we have added in the revised version of the manuscript: “This study was conducted according to the guidelines of the Declaration of Helsinki and approved by the Ethics Committee of the Hospital 12 de Octubre of Madrid (N° CEI 18/221)”.

  1. I was wondering if there is any bias in the way the specimens were collected. This should be a little more descriptive here. Is there a bias toward certain age groups? Also, wouldn't the values vary considerably depending on whether the person has had a severe infection or not? I would be curious to know if such people are separated. Of course, the sample of people is limited, so it may not be possible to go that far, but in that case, I think it should be described as a limitation of this paper.

We thank the reviewer for the comment and understand his/her question. In fact, we followed a series of exclusion criteria when choosing study participants, which information has been added in the revised version of the manuscript: "... The exclusion criteria were as follows: 1) severe and unstable medical conditions, or a history of chronic diseases; 2) psychiatric comorbidity; 3) taking medications, such as anti-inflammatory agents, muscle relaxants, corticoids, and antidepressants; 4) previous surgery; 5) pregnancy; 6) recent infections; and 7) non-cooperation during the evaluation.”

  1. The authors use the notation P=0.05 in the text, but a significant difference usually occurs when P is less than a specified value, such as 0.05>P, so this notation could be seen as no different. The statistics section fits the authors' definition, but it differs from the definitions in other papers, making me feel uncomfortable. I would like them to explain the justification for this definition.

We thank the reviewer for notifying us of this error, since a p-value≤0.05 is considered statistically significant in this paper, not higher.

  1. I don't deny the possibility that P=0.05, but it has been mentioned many times in this paper. I think the probability is low, and I don't mean to be rude, but I would appreciate your scrutiny to see if this is not some kind of error.

We understand the doubts of the reviewer, and the p-values have been checked carefully.

  1. For the purpose of this issue, is it necessary to look at women and men inclusively? I thought the values might be quite different by gender. A close examination of the paper reveals a mention of Gender in Lines 246-247, but this may be an overstatement since there are no data analyzed separately for men and women.

We thank the reviewer for the comment, and we agree that there are usually clear differences between men and women in their oxidative and immune states. However, when we analyzed our data, in terms of ImmunolAge (biological age estimated based on the immune function of the participants), we did not find significant differences between men and women; both groups followed the same dynamics regardless of their sex and, thus, no results are shown referring to sex differences in the paper because they do not seem relevant. In this line, our group observed sex differences when creating the Immunity Clock to calculate biological age, in which sex was added as an independent variable for the construction of the model; but it was not included in this paper because the selected immune variables explained the differences more than sex.

  1. Many figures have many samples with vertical axis values close to 0. Can they rule out the possibility that these are not measured due to sample problems? If there are criteria for evaluation, they should be stated in the method.

We understand the doubts of the reviewer. We understand that it may seem that getting values close to zero is an error or a sampling problem, but we can verify that it is not. They are the normal physiological concentrations of these compounds, which can have values from close to zero to a few tens, depending on the person, the state of each component of the glutathione cycle, and the type of samples used. Below, we list a series of publications that corroborate this.

-Martínez de Toda I, Vida C, Garrido A, De la Fuente M. Redox Parameters as Markers of the Rate of Aging and Predictors of Life Span. J Gerontol A Biol Sci Med Sci. 2020;75(4):613-620. doi:10.1093/gerona/glz033

-Vida C, Kobayashi H, Garrido A, et al. Lymphoproliferation Impairment and Oxidative Stress in Blood Cells from Early Parkinson's Disease Patients. Int J Mol Sci. 2019;20(3):771. Published 2019 Feb 12. doi:10.3390/ijms20030771

-Martínez de Toda I, Miguélez L, Siboni L, Vida C, De la Fuente M. High perceived stress in women is linked to oxidation, inflammation and immunosenescence. Biogerontology. 2019;20(6):823-835. doi:10.1007/s10522-019-09829-y

-Martínez de Toda I, Miguélez L, Vida C, Carro E, De la Fuente M. Altered Redox State in Whole Blood Cells from Patients with Mild Cognitive Impairment and Alzheimer's Disease. J Alzheimers Dis. 2019;71(1):153-163. doi:10.3233/JAD-190198

-Díaz-Del Cerro E, Vida C, Martínez de Toda I, Félix J, De la Fuente M. The use of a bed with an insulating system of electromagnetic fields improves immune function, redox and inflammatory states, and decrease the rate of aging. Environ Health. 2020;19(1):118. Published 2020 Nov 23. doi:10.1186/s12940-020-00674-y

  1. There are two H's in Figure 1.

We thank the reviewer for notifying us of this mistake. Figure 1 has been corrected in the revised manuscript.

  1. If I am not mistaken, the explanations for Fig. 1 and Fig. 2 are mixed in 3.1 or there is no explanation for Fig. 2 in the text? Clarifying whether the description refers to Figure 1 or Figure 2 would increase readability.

We understand the confusion of the reviewer; therefore, we have separated the explanation of Figures 1 and 2 into two different paragraphs.

Round 2

Reviewer 2 Report

Cerro et al. present here an updated version of their study concerning the thiol-based redox status in cellular and plasma blood fractions from aged individuals. Clearly, the authors present now a more consistent contribution for Antioxidants/MDPI readers, as most of the previous recommendations and suggestions from reviewers was apparently considered. However, I still have some minor suggestions for the authors before the potential acceptance of this study:  

(i) Graphical Abstract:

(i1) Please, reorder the presentation of the blood fractions in the figure, please! Actually, from the "whole blood" you isolated the "plasma" and the "cellular (white and red) fractions". Then, from the "white fraction", you isolated neutrophils and lymphocytes, whereas from the "red fraction", you obtained the erythrocytes.

(i2) Although H2O2 is also a substrate for GPX, this enzyme preferably uses lipid hydroperoxides as major substrates (LOOH or ROOH), which are byproducts of lipid peroxidation chain reactions. Therefore, this part of the figure should be improved in this sense, please.

(i3) based on the Free Radical Theory of Aging, the scale of antioxidants & prooxidants should not be balanced, since prooxidant events always prevail over AO defenses. Please, fix the figure.

(ii) Please, reformulate the Immunity Clock mathematical formula to present a single mathematical expression! Use symbols to designate the different variables (similar to the GPX expression, few lines below, lines 217-220)

(iii) As the authors probably know, there are different ways to express the so-called "reducing power"of a bilogical matrix, based  on GSh and GSSG contents. The simple ratio between GSH/GSSG is one of them, but, more accurate approaches were observed when the 1:2 stoichiometry of GSH to GSSG is considered. For example:

Reducing Power = [GSH]/(2*[GSSG] + [GSH]), where [GSH] is the concentration of reduced glutathione, and [GSSG} is the concentration of oxidized glutathione in samples.

I presume it is worthy trying to recalculate the ImmunityClock x Thiols correlations also following this expression. 

English is fine but it is always recommended to double, triple check the MS before acceptance or publication.

Author Response

Reviewer 2: Cerro et al. present here an updated version of their study concerning the thiol-based redox status in cellular and plasma blood fractions from aged individuals. Clearly, the authors present now a more consistent contribution for Antioxidants/MDPI readers, as most of the previous recommendations and suggestions from reviewers were apparently considered. However, I still have some minor suggestions for the authors before the potential acceptance of this study:  

(i) Graphical Abstract:

(i1) Please, reorder the presentation of the blood fractions in the figure, please! From the "whole blood" you isolated the "plasma" and the "cellular (white and red) fractions". Then, you isolated neutrophils and lymphocytes from the "white fraction", whereas from the "red fraction", you obtained the erythrocytes.

(i2) Although H2O2 is also a substrate for GPX, this enzyme preferably uses lipid hydroperoxides as major substrates (LOOH or ROOH), which are byproducts of lipid peroxidation chain reactions. Therefore, this part of the figure should be improved in this sense, please.

(i3) based on the Free Radical Theory of Aging, the scale of antioxidants & prooxidants should not be balanced, since prooxidant events always prevail over AO defenses. Please, fix the figure.

We thank the suggestions of the Reviewer, which have been considered to improve the graphical abstract.

(ii) Please, reformulate the Immunity Clock mathematical formula to present a single mathematical expression! Use symbols to designate the different variables (similar to the GPX expression, few lines below, lines 217-220)

Following the suggestion of the Reviewer, the mathematical formula of the Immunity Clock has been written in a single mathematical expression to facilitate its reading and understanding

 (iii) As the authors probably know, there are different ways to express the so-called "reducing power" of a biological matrix, based on GSh and GSSG contents. The simple ratio between GSH/GSSG is one of them, but, more accurate approaches were observed when the 1:2 stoichiometry of GSH to GSSG is considered. For example:

Reducing Power = [GSH]/(2*[GSSG] + [GSH]), where [GSH] is the concentration of reduced glutathione, and [GSSG} is the concentration of oxidized glutathione in samples.

I presume it is worthy trying to recalculate the ImmunityClock x Thiols correlations also following this expression. 

We agree with the Reviewer that there are multiple ways to express the so-called "reducing power” of a biological matrix, based on GSH and GSSG contents. However, we have validated the GSSG/GSH ratio as a marker of longevity and biological age; therefore, we have used this way, and no other, to study the "reducing power" in relation to the Immunity Clock.

Martínez de Toda, I., Vida, C., Garrido, A., & De la Fuente, M. Redox Parameters as Markers of the Rate of Aging and Predictors of Life Span. The journals of gerontology. Series A, Biological sciences and medical sciences 2020, 75(4), 613–620.

Martínez de Toda, I., Vida, C., Sanz San Miguel, L., De la Fuente, M. When will my mouse die? Life span prediction based on immune function, redox and behavioural parameters in female mice at the adult age. Mech Ageing Dev 2019a, 182:111125.

Martínez de Toda, I., Vida, C., Sanz San Miguel, L., De la Fuente, M. Function, Oxidative, and Inflammatory Stress Parameters in Immune Cells as Predictive Markers of Lifespan throughout Aging. Oxid Med Cell Longev 2019b, 2019:4574276.

Reviewer 3 Report

My concerns are mostly addressed in this version. I consider the paper to be interesting and informative.

One point, I thought it would be better to correct the inconsistency in the number of digits in the notation of significant differences, etc.

Author Response

Reviewer 3: My concerns are mostly addressed in this version. I consider the paper to be interesting and informative.

One point, I thought it would be better to correct the inconsistency in the number of digits in the notation of significant differences, etc.

We appreciate the Reviewer's comment. In the new revised version of the manuscript you can see that we have unified the number of digits.
